# A Systematic Literature Review of Factors Affecting the Timing of Menarche: The Potential for Climate Change to Impact Women’s Health

**DOI:** 10.3390/ijerph17051703

**Published:** 2020-03-05

**Authors:** Silvia P. Canelón, Mary Regina Boland

**Affiliations:** 1Department of Biostatistics, Epidemiology and Informatics, Perelman School of Medicine, University of Pennsylvania, Philadelphia, PA 19104, USA ; silvia.canelon@pennmedicine.upenn.edu; 2Institute for Biomedical Informatics, University of Pennsylvania, Philadelphia, PA 19104, USA; 3Center for Excellence in Environmental Toxicology, University of Pennsylvania, Philadelphia, PA 19104, USA; 4Department of Biomedical and Health Informatics, Children’s Hospital of Philadelphia, Philadelphia, PA 19104, USA

**Keywords:** climate change, timing of menarche, women’s health

## Abstract

Menarche is the first occurrence of a woman’s menstruation, an event that symbolizes reproductive capacity and the transition from childhood into womanhood. The global average age for menarche is 12 years and this has been declining in recent years. Many factors that affect the timing menarche in girls could be affected by climate change. A systematic literature review was performed regarding the timing of menarche and four publication databases were interrogated: EMBASE, SCOPUS, PubMed, and Cochrane Reviews. Themes were identified from 112 articles and related to environmental causes of perturbations in menarche (either early or late), disease causes and consequences of perturbations, and social causes and consequences. Research from climatology was incorporated to describe how climate change events, including increased hurricanes, avalanches/mudslides/landslides, and extreme weather events could alter the age of menarche by disrupting food availability or via increased toxin/pollutant release. Overall, our review revealed that these perturbations in the timing of menarche are likely to increase the disease burden for women in four key areas: mental health, fertility-related conditions, cardiovascular disease, and bone health. In summary, the climate does have the potential to impact women’s health through perturbation in the timing of menarche and this, in turn, will affect women’s risk of disease in future.

## 1. Introduction

Menarche is the first occurrence of a woman’s menstruation, an event that symbolizes reproductive capacity, and which in many cultures represents a woman’s transition from childhood into womanhood [1]. During this transition, a woman undergoes many cultural, sociological, psychological, and physical changes [1,2]. The “timing of menarche” is the age when a young woman first begins menses, and is often studied in terms of describing those with early or late menarche relative to a certain population. Timing of menarche is important to women’s health in general as it has been associated with a country’s fertility and female mortality rate; specifically, menarche occurs later in countries with high mortality and fertility rates [3]. Studies have shown that the average age of menarche has declined in industrialized countries [4,5,6,7,8,9,10,11,12,13] in recent history. Some of these studies have been anecdotal in nature [5], while others have been more systematic [4,6,7,8,9,10,11,12,13].

Decreases in mean age of menarche among industrialized nations appear to be somewhat related to similarly observed increases in height (with the hypothesis being that adequate nutrition increases height and decreases age of menarche); however, this does not fully explain the relationship [6]. Significant variance in mean age of menarche also exists based on country, suggesting the potential for genetic and/or environmental causes [14]. Decreases in average age of menarche have been observed in developing countries as well, where the causes are often difficult to ascertain [14]. Regarding resource-rich countries, researchers have proposed the theory that the rising obesity epidemic may be the reason for the decline in age of menarche observed in industrialized nations such as the United States of America (US) [15]. However, one study appeared to refute this theory and found that population-level shifts in body mass index (BMI) and age of menarche were independent, but sometimes coincident processes [15]. Studies in other industrialized nations such as South Korea, where obesity rates are relatively low, also report strong negative correlations between BMI and age of menarche [7,8]. Another study found that the reduction in average age of menarche in Germany from 18 to 12–13 years, which has been occurring since the 1800s, was not driven solely by the nutritional, health, and economic changes that occurred [16]. Therefore, the cause of the population-level reduction in age of menarche that has been observed in industrialized countries remains unknown.

When considering socioeconomic status as a factor, a recent US study found income status to be related to timing of menarche over time when the proportion of girls experiencing early menarche (<11 years old) increased over a 50 year time period only for those girls of a low socioeconomic position [10]. In contrast, another recent study conducted in the United Kingdom (UK) spanning an 85 year time period found a decline in the timing of menarche across all socioeconomic groups [12]. Therefore, socioeconomic status alone is unlikely to explain the changes in timing of menarche over time.

Studies of menarcheal trends have focused mainly on single factors. Some are present at the individual level, such as physical variables like BMI and height [17], and environmental variables like socioeconomic status. Other single-factor studies have evaluated factors present at a broader scale, using climate variables like regional temperature [18,19] and altitude [20]. One recent study demonstrated the potential for interdependence between these types of factors when assessing female fecundity [21], opening the possibility for a similar dynamic to exist in other outcomes related to reproduction, such as menarche. There is limited literature to suggest an underlying mechanism for the changes seen in timing of menarche over time. This emphasizes a need for a comprehensive summary of individual- and broad-scale effects on women’s health. Furthermore, climate change over time could be reasonably projected to influence timing of menarche, adding another layer of complexity and opportunity for study. For the purposes of this review, “climate change” is defined as any changes in the climate that have occurred, focusing specifically on changes occurring within the past hundred years. Our definition is agnostic to the causes of climate change and whether or not climate change events are man-made (i.e., anthropogenic) [22] or purely the evolution of geologic age [23].

Our systematic literature review focused on both causes and consequences of abnormal variation in the timing of menarche (either early or late). This included environmental, physiological, sociological, disease-related, and genetic contributions to menarche changes. Psychological changes were also included as either diseases or social factors, depending on the specific study. This review detailed all of the interwoven etiologies and then focused on ways that climate change could alter the timing of menarche. Our goal was to provide a comprehensive assessment of the ways that climate, and also climate change, can affect the timing of menarche, and potential consequences that this could have on women’s health.

## 2. Materials and Methods

The literature review was conducted according to PRISMA guidelines [24].

### 2.1. Systematic Review of Literature

Our literature review focused on the timing of menarche and the role of climate change in altering the timing of menarche among women. Our first step was to search for relevant literature articles pertaining to timing of menarche. Four databases were searched: PubMed, EMBASE, SCOPUS, and Cochrane. Site licenses from the University of Pennsylvania libraries were used to search SCOPUS, EMBASE, and Cochrane. The search query “timing of menarche” was used to identify relevant timing of menarche papers.

The initial query was generic and related to the timing of menarche. All of the resulting articles were manually reviewed and grouped in terms of their relevance to the climate-change-related phenomena discussed in this review. Note that the query interface within PubMed automatically maps terms to their respective medical subject headings (MeSH) terms using PubMed automatic term mapping [25]. After retrieving results from each database, duplicate studies were removed using exact PubMed ID match. Articles were manually reviewed and title, author list, and publication date were compared to further identify duplicate publications in cases where PubMed IDs were absent. Conference abstracts were removed, but conference papers were retained because conference abstracts often represent early versions of research that are later published in more complete research articles. Articles were retained if they met the following additional criteria: (1) were human studies (i.e., non-human studies were removed); (2) focused on timing of menarche with regards to causes or consequences (articles that simply used age at menarche as a proxy for puberty in an otherwise unrelated study were removed); (3) were not simply descriptive studies of a population’s typical age at menarche; and (4) were not simply mathematical modeling papers. Both authors then reviewed the final set of papers and organized them into themes. Studies were not restricted based on the number of years since publication because, in some cases, old studies were landmarks in determining certain factors that are related to either early or late menarche.

### 2.2. Identification of Themes

The articles’ abstracts were reviewed and grouped according to themes. If the article pertained to some form of environmental exposure (e.g., crop-related, toxin-related), it was labeled as pertaining to the environment. If the article involved some disease-related factor that altered timing of menarche, it was classed a disease-related article; in some cases, these were delineated further into physiology-related articles if the focus was more on a physiological aspect than a disease. Finally, if articles involved sociological aspects of timing of menarche, they were grouped as sociological. Themes were grouped into causes vs. consequences of early or late menarche. A category for genetics was included for articles investigating the genetic factors related to timing of menarche.

## 3. Results

### 3.1. Systematic Review of Literature

This review involved a search of PubMed, EMBASE, SCOPUS, and Cochrane for articles on timing of menarche. A total of 137 papers from PubMed, 163 from EMBASE, 145 from SCOPUS, and 0 Cochrane reviews were identified as of 5 December 2019. Eight trials on timing of menarche were found in Cochrane’s database; however, this review focused on published literature articles and trials were therefore excluded (although publications from those trials may appear in our results from the other database searches). An initial set of 145 papers from SCOPUS was combined with 41 additional records from EMBASE and 15 additional papers from PubMed. Duplicate studies were removed using PubMed ID (if available), title, publication date, and author list information. Non-English studies, including one in German and one in Spanish, were removed. Furthermore, 32 conference abstracts, 1 book chapter, and 1 note were excluded. Four articles were excluded for being non-human studies, including two in rhesus monkeys, one in mice, and one in *C. elegans.* Furthermore, papers that used age at menarche as a proxy for puberty status and were not exploring either causes or consequences of early/late menarche were removed, as were articles that were purely descriptive studies of a population’s typical age at menarche. Reviews were also excluded and the final review set used to identify themes contained 117 articles. The overall selection process is shown in Figure 1. Importantly, studies from both developed and developing countries were included in our final review dataset. Studies were not excluded based on country of origin.

### 3.2. Themes Identified as Important for Understanding a Woman’s Timing of Menarche

This review identified 117 distinct articles that were then grouped according to the themes identified in the articles, with some articles describing multiple themes. Figure 2 shows the number of publications per theme identified by this review. Articles were further grouped into factors (e.g., diseases, environmental exposures) by whether the study found an association with early or late menarche. After carefully reviewing the 117 papers and categorizing them into themes (Figure 2), five articles that were not relevant were excluded. One excluded study was a drug outcome study for treating early menarche, which was not the focus of this review [26]. Two studies were excluded that described physiological changes that occur due to the onset of menarche in general, but with no discussion about early vs. late menarche [27,28], along with another focused on visualizing the changes of menarche for patients [29]. A dubious study on physical attractiveness was also excluded [30]. A detailed analysis was performed on all remaining 112 studies. The reported effects of environmental causes, disease causes and consequences, and social causes and consequences on early vs. late menarche are shown in Figure 3.

#### 3.2.1. Environmental Causes

##### Growth/Nutrition

Environmental causes of early menarche have been more frequently studied than those associated with late menarche, with 15 articles reporting findings related to early menarche vs. 7 for late menarche and 3 studying both early and late menarche. The most commonly studied environmental causes were related to nutrition, adequate food intake, and proper growth parameters (e.g., height, weight) [17,31,32,33,34]. One study noted that greater childhood growth coupled with high socioeconomic status was associated with in early menarche [35]. Nutritional factors that were found to be associated with early menarche included high protein intake [36], high leptin [37,38], and soy formula [39]. While the relationship between soy formula usage and early menarche was found in one study [39], it was not found in another study, suggesting that this relationship may warrant further study [40]. The variables associated with late menarche included high vegetable intake [36], flavonol intake [41], and food insecurity [42]. Another study found that nutritional geometry and lower dietary protein (relative to carbohydrates and fats) predicted early puberty and timing of menarche, suggesting that non-protein macronutrients are also important [43]. This seems to contradict earlier studies that suggested that high protein intake was associated with early menarche [36], indicating that perhaps it is the relative balance of nutrients (protein vs. carbohydrates and fats) [43] that ultimately plays a role in timing of menarche. Milk fortified with calcium or calcium and vitamin D did not appear to alter the timing of menarche [44].

Closely related to nutrition is proper growth. If inadequate growth is evident then menarche is delayed, and proper growth depends on nutrition. Menarche occurred earlier among those who had achieved adequate growth indicators, including height [31,32,33], weight [32,33,34], vital capacity [32], and arm circumference [33]. Conversely, a study from Bangladesh found that those with physical stunting (thought to be due to inadequate growth from inadequate nutrition) had later menarche than those without stunting [45]. Another study found that the significant effect of size at birth on timing of menarche became insignificant after adjusting for growth between 2 and 7 years of age, indicating that growth between these ages plays a strong role in the timing of menarche and early life factors may be mediated by growth later in childhood [46]. A related study found that girls who were heavier at 9 years old had an early menarche, as did those born to younger mothers and those born prematurely, indicating some interdependency between early developmental factors, weight during childhood, and timing of menarche [47]. Another study pointing to the effect of in utero exposure was conducted on girls adopted from China and raised in the United States of America; these girls went through menarche at an age similar to those born and raised in China, suggesting either a genetic contribution to age of menarche or an in utero exposure [48].

Another study found a U-shaped distribution between weight and later fertility, with fat determining the time of menarche, indicating that those on the extremes (very underweight or very overweight) are at risk for fertility issues and potentially alterations in timing of menarche [49]. A study on female fertility at the country level using birth rates and average country statistics on body mass index (BMI) found that a country’s BMI was a strong predictor for a country’s fertility rate [21].

##### Toxins and Pollutants

Several non-nutrition studies have pointed to a role of environmental pollutants and toxins in altering the timing of menarche. Exposure to lead [50], polychlorinated biphenyl (PCB) [51], and in utero exposure to polybrominated diphenyl ether (PBDE) [52] were all associated with late menarche. Those exposed to zearalenone [53] were significantly shorter in stature, with a non-significant trend toward later menarche [53]. Phthalate exposure resulted in either early or late menarche depending on the specific species of phthalate [54]. Increases in di(2-ethylhexyl) phthalate were associated with later menarche, whereas increases in 2,5-dichlorophenol and benzophenone-3 were associated with early menarche [54]. Similarly, radiation exposure resulted in either early [55,56] or late [56] menarche, depending on the study. One study that found early menarche was among radiation-exposed individuals being treated for acute lymphoblastic leukemia [55]. A study involving radiation exposure via cranial radiotherapy for central nervous system tumors found both early and late menarche among those exposed [56]. Another study of brain cancer patients found indicators of ovarian dysfunction with no apparent effect on timing of menarche (study size = 21 patients) [57], indicating that the relationship between radiation exposure due to cancer treatment and perturbations in timing of menarche is likely multifactorial. It is possible that the effect may depend on the location radiated, dosage, and length of exposure, which is likely cancer-specific. Another study investigating the effects of breast cancer incidence and atomic bomb survivors in Japan and its relationship to timing of menarche found that the breast cancer incidence due to radiation was high regardless of the stage of the breast’s differentiation, indicating that radiation exposure itself maybe a breast carcinogen [58]. Exposure to endocrine-disrupting chemicals [59] and hair products [59], and in utero exposure to atrazine [60] were all associated with early menarche (Figure 3A). On the other hand, in utero exposure to organochlorine did not appear to alter the timing of menarche [61].

#### 3.2.2. Disease Causes and Consequences

##### Diseases that Affect Timing of Menarche (Causes)

Literature indicating potential disease-related causes of early or late menarche is presented in this section (Figure 3B). Most of these studies were association studies (and not true causal studies); however, they point to pre-existing disease states that likely alter the underlying physiology, potentially resulting in perturbations in the timing of menarche.

Disease causes for early menarche include myelomingocele (a type of spina bifida) [62], fat mass [63], inverse association with BMI (i.e., heavier BMI—earlier menarche) [64,65], metabolic syndrome [66], and insulin resistance [67]. One study linked bulimia with early menarche [68]. Bulimia is an eating disorder with genetic, environmental psychological, and social links, but, for the purposes of this review, bulimia was categorized with the diseases for simplicity. Early menarche was not associated with birth weight [69], pre-term birth [70], being small for gestational age [71], or congenital adrenal hyperplasia [72]. As indicated in the environmental causes section, these early life risk factors (birth weight, pre-term birth, small size for gestational age) do not appear to be the main drivers for early menarche. One study found that expected birth weight coupled with higher BMI during childhood increased the risk of early menarche [73]. Tam et al. found that menarche was earlier in girls who were born long and light and who had higher fat mass and IGF-1 during childhood [74]. These two studies suggest the possibility of a linked relationship between birth weight and BMI, weight gain, and timing of menarche. Another study by Winterer et al. found that an increase in the brain-to-body ratio delayed menarche—suggested to be due to increased caloric burning in brain development that delays the onset of menses [75]. Late menarche was associated with type 1 diabetes [64], anorexia [76], scoliosis [77], Turner syndrome (one X chromosome) [78], polycystic ovarian syndrome (PCOS) [79], and immune-related diseases, including juvenile rheumatoid arthritis [80,81], and Crohn’s disease [82].

##### Diseases that Result from Perturbations in Timing of Menarche (Consequences)

Disease consequences are diseases resulting later in life (i.e., at a time after menarche) that were correlated with timing of menarche (Figure 3C). More diseases were correlated with early menarche than late menarche. Diseases/conditions associated with late menarche include osteoporosis [83] and two fertility-related conditions, including fetal loss [84] and longer time to first baby [84], and hirsutism (excessive body hair growth) [85]. Diseases/conditions associated with early menarche include short stature [86], elevated BMI [87], fertility-related diseases/conditions including increased antral follicle count (typically a sign of increased fertility) [87], premature menopause [88], hysterectomy [89], and mental health conditions including depression [90], thoughts of self-harm (but not suicidality) [91], fear [92], distress [92], externalizing disorders [92], and behavioral problems [93]. Short stature is more a characteristic of the patient state and can indicate lack of adequate nutritional intake [86]. However, for purposes of this review, short stature was characterized as a disease/condition [86]. Cardiovascular disease [94], including increased carotid artery intima-media thickness (an indicator of cardiac age and a risk factor for cardiovascular disease) [95] was associated with early menarche. Another study found that either early or late menarche was associated with higher risk of adverse cardiovascular disease outcomes [96], perhaps indicating that being either early or late with regards to menarche puts a strain on cardiovascular development.

Early menarche was associated with breast cancer [97,98]. It appears that the link between breast cancer and early menarche has been established for some time [98]. However, one study found that it is the time interval between age at menarche and age at first birth that was associated with increased risk of certain hormonally sensitive breast cancers (lobular and hormone-receptor-positive tumors), particularly among white women [97]. The authors posited that this time interval between age at menarche and age at first birth explained the relationship between breast cancer and age at menarche [97]. Another study investigated the socioeconomic differences between white and black women and age at menarche (given that there is a disparity among breast cancer risk between the two groups). They found that increased household income resulted in a protective effect against early menarche for white girls, whereas black girls were at increased risk of early menarche even in the same elevated household income group [99]. Another study found that both early and late age at menarche could be predictive of certain breast cancers, but that it was also related to parity (i.e., number of births a woman experiences over her life course), indicating that the relationship between age at menarche and breast cancer risk is complex [100].

Disease causes and consequences are summarized in Table 1.

#### 3.2.3. Social Causes and Consequences

##### Social Factors that Affect Timing of Menarche (Causes)

Many social factors have been identified as potentially causing early age at menarche, and a few have been identified as causing late menarche (Figure 3D). A study conducted using data from Poland during the Soviet Union era (1950s–1970s) found that high socioeconomic status was linked with early menarche [101] and low socioeconomic status was linked with late menarche [101]. This appears contradictory to a study by Braithwaite et al. that found that white girls of high socioeconomic status were protected against early menarche [99]. However, this Braithwaite et al. study also found that black girls did not experience the same protective effect even when they belonged to a high socioeconomic status group [99]. Therefore, it is possible there is some relationship between high socioeconomic status and low stress that is protective against early menarche, given that it is likely that the white Polish girls in the Soviet Union were under high levels of stress [101], distinguishing them from the white girls in the Braithwaite et al. study [99].

Lack of paternal influence and quality of the paternal relationship during early childhood has been linked with early menarche [102,103,104,105,106]. One study found that the first five years of life were the most critical for paternal influence to be protective against early menarche [104]. Another study found that it was not paternal absence, but the lack of a quality relationship with one’s father that was linked with early menarche [106]. Depression, early childhood paternal absence, and early age at menarche were related to each other in another study [105].

Another factor linked with early menarche is stress. One study investigated different types of stress and found that stress due to harshness, but not unpredictability, resulted in early menarche [107]. Another study found that both age at menarche and C-reactive protein (a biological indicator of inflammation and stress) were negative predictors of estradiol [108]. This indicates that physiological changes in stress (i.e., elevated C-reactive protein) could affect timing of menarche.

Another study found that sharing a room with male family members increased the age at menarche [109], and that having more siblings also increased the age at menarche [109]. This could indicate that there is something about proximity to males that delays menarche. Other social causes of early menarche include lack of athleticism in girls [110]. Another study found that swimmers, track athletes, and rowers were no different to controls with regards to age at menarche (i.e., neither early or late) [111]. Therefore, lack of athleticism appears to result in early menarche [110] while specific sports appear to result in typical age of menarche [111]. However, a Korean study reported conflicting results, with early and late menarche observed among girls participating in physical activity and normal menarche observed for those with sedentary activity [112]. Therefore, it is possible that the specific type of physical activity could affect timing of menarche, but likely not physical exercise in general, given the conflicting results.

##### Social Factors that Result from Perturbations in Timing of Menarche (Consequences)

Social consequences of early or late menarche are social effects that occur later in life (i.e., at a time after menarche) that have been correlated with timing of menarche (Figure 3E). Most of these social consequences have been correlated with early menarche. These include fertility consequences such as early first pregnancy [113] and more offspring [114], which have also been connected with late menarche [114], although this link may be mediated by elevated BMI and obesity [114].

Several mental health consequences are associated with early menarche, including depression [115,116], eating disorders [117], and body image issues [117]. Depression, eating disorders, and body image issues could be considered disease consequences, but in many cases authors point to the social consequence of having ones first menses early relative to their peers [118], resulting in inconvenience, ambivalence, and confusion [118], and that these factors may elevate their risk for depression. Post-menarche puberty status among girls was associated with increased depression symptoms among white girls, but this was not observed among African-American and Hispanic girls [119]. This suggests a social component to these phenomena. An elevated risk for depression was observed among those with early menarche at 13 and 14 years old [116]. Another study found that it was not early menarche so much as early breast development that increased depressive symptoms among teenage girls [120]. This further suggests that the social consequences of early menarche, including early puberty and early breast development, may be contributing to increased depressive symptoms in teenage girls. Another study found that a woman’s subjective view regarding whether or not her menarche was early or late was more associated with perceived social experience of a positive or negative menarche than the actual reality of whether the age of menarche was early or late [121]. Anxiety related to timing of menarche was elevated among those with early menarche, and also among those with a history of sexual abuse [2].

Other social consequence studies included a study that found that girls with early menarche were at increased risk of alcohol abuse [122]. Perception of gender norms with regards to traditional gender roles and gender bias in allocation of resources was not associated with age at menarche [123].

#### 3.2.4. Genetics and Other Factors

Approximately half of the phenotypic variation among girls from developed nations appears to be due to genetic and heritable factors [124,125,126], with differences observed between races and ethnicities [127]. Studies have associated the sex-hormone-binding globulin (*SHBG*) gene with timing of menarche, with girls with longer TAAAA allele repeats having late menarche [128]. Another study found that a single nucleotide polymorphism (SNP) in the leptin receptor gene (*LEPR*) resulted in late menarche among Korean women [129]. Two loci were identified as suggestive of age of menarche in another study of Chinese and Korean women [130]. A few smaller scale studies failed to find any genetic markers for age at menarche [131,132,133,134], and another reported findings suggestive of an interaction effect between multiple SNPs and age at menarche [135]. However, a recent study found hundreds of variants associated with age at menarche, including two rare mutations in *MKRN3* and *DLK1* genes that were paternally inherited [136]. More work is still needed to understand the role of genetics on timing of menarche, including whether there are combined gene–environment interactions that play a role in altering the timing of menarche.

### 3.3. Potential for Climate Change to Impact Timing of Menarche

Climate change could impact the timing of menarche in several ways. In particular, it could result in increases in hurricanes, twisters, avalanches, and other extreme weather events. This could then result in increased release of toxins in the environment. Several toxins are known to perturb timing of menarche (Figure 3A). In addition, weather events could alter crop availability and intake of vegetables and protein, which is known to affect timing of menarche. These perturbations in timing of menarche could then result in increased disease burden with respect to increases in bone health conditions, cardiovascular diseases, mental health diseases, and fertility-related conditions. A conceptual schema detailing the impact that climate change could have on disease burden via perturbation in timing of menarche is shown in Figure 4.

## 4. Discussion

This review described the disease causes and consequences of early and late menarche. The environmental causes and social causes and consequences of early and late menarche were explored. Many of these causes of perturbation in menarche could be altered and affected by climate variables. Crop and food availability, which is required for appropriate onset of menarche, is heavily impacted by our climate and is sensitive to climate changes. In addition, toxin and pollutant release could be affected by hurricanes and other extreme weather events. The nuances of these impacts are discussed below.

### 4.1. Impact of Crops and Crop Production on Timing of Menarche

Reproductive events, including the timing of menarche, are facilitated or limited by the availability of energy required to perform necessary functions. With the understanding that adipose tissue stores energy derived from food sources to later carry out a variety of metabolic processes, it follows that body fat availability would influence reproductive functions [137,138]. In fact, menarche has been found to be inversely related to levels of leptin, a hormone produced by fat cells [38]. In 2007, Lassek and Gaulian studied fat distribution and skeletal maturity as factors in menarche and determined that menarche is more closely tied to fat distribution and inversely related to leptin levels [139]. Earlier age at menarche was correlated with higher fat intake [140] and BMI [141,142], and higher consumption of animal (as opposed to vegetable) protein [143].

Likewise, energetic challenges can delay menarche and suppress reproduction due to inadequate food intake and/or excessive physical exertion without a compensatory increase in food intake. When considering physical exertion, Chavarro et al. found age at menarche to be positively associated with two or more hours of daily physical activity [144], and Merzenich et al. found a delay in menarche with increased sports activity [140]. These relationships are magnified among highly trained athletes, with greater energy expenditures. These include professional ballet dancers [145,146], gymnasts, and long-distance runners [147,148,149]. Chronic malnutrition has also been found to delay menarche, specifically among those living in developing areas of India [150] and the Chimba province of Papa New Guinea [151]. More broadly, the effect of these types of energetic challenges on other reproductive events such as ovulation has been well studied. Elias et al. studied effects of the 1944–1965 Dutch famine on reproductive function, and found that girls exposed to severe famine before menarche were roughly 1.5 times more likely to have irregular menstrual patterns for a longer period of time relative to those unexposed. Those exposed to severe famine 2 or more years after menarche were almost 9 times more likely to experience prolonged menstrual irregularity than those unexposed [152]. Therefore, it is not surprising to find that energetic challenges affect timing of menarche as well.

Adequate food intake and balanced energy expenditure, both critical factors in reproductive function, can be limited by crop availability and, therefore, by climate conditions favorable to crop production. This interdependence creates a dynamic, particularly in subsistence societies, where seasonal changes in climate conditions cause seasonal changes in nutrition intake and food-related energy expenditure, leading to seasonal variations in ovulation [153,154,155], specifically calorie-based amenorrhea. During the dry season in arid northwest Kenya, the Turkana pastoralists who almost exclusively rely on herding for food (e.g., milk, meat) must move herds as needed in search for water and vegetation. Coupled with the water hauling often done by the Turkana women, this creates a climate-driven energetic challenge that could explain the seasonal patterns of ovulation observed in this society [156]. Seasonal patterns in ovulation and conception have also been documented among the Lese farmers in Zaire [157,158] and the San foragers of Botswana and Namibia [159,160,161], respectively. These observations have been attributed to the associated seasonality of rainfall and food availability. Interestingly, while the agro-pastoralist women of the Himalayan foothills do not experience seasonal variability in food availability, a seasonal variation in ovulation is still observed due to the seasonal variation in expenditure of energy, which is required in excess during the monsoon season and not as much during the winter [162].

Seasonality in menarche has also been described in multiple populations, though the connection with seasonal variation in climate conditions is not as clear, as these populations are not considered subsistence societies. Peak menarche frequencies have been found in January and July among girls in the United States [163], north Italy [164], and Norway [165]. Peaks in April and October have been described among girls in Thailand [166] while in Japan girls experience peak menarche in April and August [167]. These populations may experience seasonal changes in energetic challenges due to economy- or culture-driven variations in food availability that influence food intake behavior rather than directly from changes in climate conditions.

### 4.2. Overview of Climate Change

The changing facets of our climate vary based on the specific location in the world. However, specific factors have been noted, including warming ocean temperatures [168,169], which is believed to be the culprit behind increases in air temperature [170,171] and humidity [171], and also changes in phenology trends (i.e., seasonal trends such as spring bloom for flowers) [170]. Warming ocean temperatures have been linked to increases in both the number of hurricanes and their severity [172]. Increases in land/mudslides and avalanches have also occurred recently, which may be due to warming temperatures [173] (this is complex to study due to increases in urbanization that can also impact soil stability). These human-factor changes (e.g., increased urbanization, roads) are thought to be equal to or greater than the effects of climatic change on land/mudslides and avalanches [174]. However, there does appear to be a significant climatic contribution to these events [174].

### 4.3. Impact of Climate Change on Pollutant Release and Timing of Menarche

Increases in hurricanes and hurricane severity, extreme weather events (cyclones), twisters, avalanches, and mud/landslides have been observed in the past 50–100 years. These factors can increase the risk of pollutant/toxin release into the environment, which in turn can alter timing of menarche (Figure 4). Hurricane Katrina was a major event that resulted in catastrophic flooding in the region of New Orleans in 2005. The flooding that resulted from the hurricane caused at least five oil spills and citywide sewage breakdowns and other waste disposal issues [175]. The water was not as toxic as initially suspected, although the sediment left behind when the floodwaters receded could result in dangerous air pollution [175]. Oil spills often result in release of many toxins into the environment, which could potentially be detrimental to age of menarche. A study found that endocrine-disrupting chemicals (EDCs) were found in the surface water near an oil and gas wastewater disposal site [176]. McDonald et al. found that increases in EDCs have been implicated in early menarche [59]. However, their study investigated EDCs in hair oils and hair products [59], which are likely somewhat different from the EDCs found in oil and gas wastewater. However, if the mechanism affecting age at menarche is the EDCs as proposed, then the specific chemical should not matter as long as endocrine disruption is taking place [177].

Hurricanes also result in increased precipitation (i.e., rainfall). Certain harmful chemicals have been found in precipitation, including polybrominated diphenyl ethers (PBDEs) [178]. PBDEs are chemicals typically found in flame-retardants. Harley et al. found that in utero exposure to PBDEs [52] was associated with late menarche. In addition, PBDEs have slight EDC properties as well, and could influence age of menarche through an EDC pathway [179]. Therefore, if PBDEs are released through precipitation (rainfall) and precipitation increases due to hurricane activity in a given region, it could be possible for PBDE exposure to increase during a hurricane. This could thus affect the timing of menarche, resulting in late menarche for girls that are exposed in utero, provided their mothers were pregnant during a hurricane and exposed to PBDE-containing rainfall.

The Hudson River (located in the state of New York) is severely contaminated with polychlorinated biphenyls (PCBs), and exposure increases when Hudson River sediment is re-suspended [180]. This demonstrates that contaminated soil sediment could affect individuals later due to release of the sediment into the water. These types of releases are more likely to occur following a hurricane. A study by Balluz et al. found that environmental pesticides were found in adolescents subsequent to hurricane Mitch in the Honduras [181]. These chemicals had not been used in 15 years, indicating the potential for re-exposure to old contaminated sediment following hurricanes as one hypothesis [181]. PCB exposure was found to cause late menarche [51] and therefore exposure to PCBs, even if via sediment release many years after the primary exposure, could result in perturbations to menarche. Many chemicals can result in either early or late menarche (Figure 3A) and, therefore, exposure due to climate changes could be detrimental to females due to exposure via contaminated precipitation or soil sediment re-release.

### 4.4. Potential for Climate Change to Alter Weather Events and Food/Crop Availability

Extensive work by the International Food Policy Research Institute [182,183] has predicted that climate change will adversely impact crop yields, producing downstream effects on food prices, production, and consumption. Specifically, rising temperatures, changes in rainfall patterns, and associated changes in water irrigation practices are expected to result in yield declines in the most important agricultural crops: wheat, maize, soybeans, and rice [182]. Price increases in these cereal grain crops are predicted to result in higher prices in animal feed and, therefore, higher meat prices, driving an overall reduction in the growth of meat consumption and an overall decline in the consumption of cereals by 2050 [182].

Past examples have demonstrated how weather related events similar to those previously described (Figure 4) impact farmers in Uganda [184]. Sweet potato farmers have reported increases in extreme rainfall, mudslides, storms, floods, and prolonged dry seasons [184]. Drought and floods had the highest impact on crop production [184], thus affecting food availability (Figure 4). Smaller farmers were more aware of the climate changes and were likely to modulate their behavior (e.g., planting drought-tolerant trees) [184]. This is likely due to their increased vulnerability to perturbations in climate.

The stability of crop production and food availability in general may be at risk due to climate change because of perturbation in supply chains [185,186]. These changes are likely to result in greater food insecurity in areas that are already vulnerable to malnutrition and under-nutrition [182,185,187]. Reductions in crop production are likely also to heavily affect farming communities that rely on farming as their main source of income [185,188], potentially causing other downstream economic issues due to lack of individual and household incomes. In fact, the World Bank reports that the majority of the global population living in poverty lives in rural areas and is employed by the agricultural sector [189]. Singh-Peterson et al. found that smaller stores were less resilient to extreme weather events, including flooding, when compared to larger more urban-based stores [190]. This further establishes that vulnerable populations are likely to be at the highest risk.

Areas of the world that have been historically subjected to highly variable climates tend to have local knowledge based on traditional customs that can be useful in combating climate change’s effects on food availability, and there is ongoing research in this space [191]. The goal would be to create an appropriate ecosystem that is robust to extreme weather events while still enabling balanced food production. However, if extreme weather events are imminent and farming practices cannot adapt appropriately, then future alterations in climate would be expected to alter the food supply chain [192] resulting in under- or over-abundance of certain foodstuffs.

Of particular importance to public health is the impact these changes could have on the consumption of nutrients found to affect the timing of menarche. Increases in protein intake [36], leptin [37,38], and potentially soy in formula [39] (was not replicated in Reference [40]) have all been implicated in early menarche. Meanwhile, high vegetable [36] and flavonol intake [41] and food insecurity [42] have been implicated in late menarche. Taken together, these studies indicate that the appropriate intake of protein, vegetables, leptin, flavonols, and soy is necessary for normal timing of menarche. Additionally, a study by Cheng et al. found that the relative balance of nutrients (protein vs. carbohydrates and fats) [43] ultimately plays an important role in timing of menarche. In summary, the ratios of these dietary food stuffs can result in either early or late menarche, which have both been linked with specific disease consequences (Figure 3C). Because alterations in the balance of these foods can perturb timing of menarche, it is probable that changes in weather events perturbing various aspects of the food supply chain could have devastating effects on timing of menarche for future generations if unaddressed. These adverse impacts are compounded among populations already vulnerable to hunger and malnutrition, due to the likelihood that food systems will experience a reduced capacity to support access to adequate food quantity and quality, and reduced economic stability, particularly as agriculture can also represent a primary source of income [187].

### 4.5. Limitations of Studies in This Review

One study from Sweden found that women’s reporting of symptoms varied by their age at menarche [193]. At age 15, fewer early maturing women (i.e., those with early menarche) reported being symptom-free than was expected by chance. However, by the time these women were 43 years old, there was no difference in their reporting of symptoms based on time of menarche [193]. This could indicate that there are potential biases in some survey-based studies that focus on asking women about their symptoms and the differences thus observed based on early vs. late menarche, as these reported symptoms may not persist into late adulthood and may only persist briefly following the onset of menses. This could affect the results of some studies that measure outcomes close to the onset of menarche. This literature review reflects the gendered language used in the articles we referenced. We use the term ‘women’, but specifically we are referring to those with menarche (first menses) and menstrual cycle experiences. We do not address gender or sexual identity or sexual orientation, as this level of detail was not captured or known in the studies referenced. In brief, the individuals included in these studies may not identify as ‘women’.

## 5. Conclusions

Collectively, the studies and analysis presented in this review shed light on the multifaceted impact that climate change could have on timing of menarche as an important marker of pubertal maturity in a woman’s reproductive lifespan. This literature review focused on 112 relevant studies on timing of menarche. The review found that climate change events, including increases in hurricanes, mud/landslides, and avalanches, and increases in extreme weather events could perturb the natural timing of menarche either through increased release of toxins and pollutants buried in soil and water or by impacting food availability via crop failure. Overall, these perturbations in timing of menarche are likely to increase the disease burden for women in four key areas: mental health, fertility-related conditions, cardiovascular disease, and bone health, as revealed in our review. In summary, the climate does have the potential to impact women’s health through perturbation in timing of menarche and this, in turn, will affect women’s future risk of disease.

## Figures and Tables

**Figure 1 ijerph-17-01703-f001:**
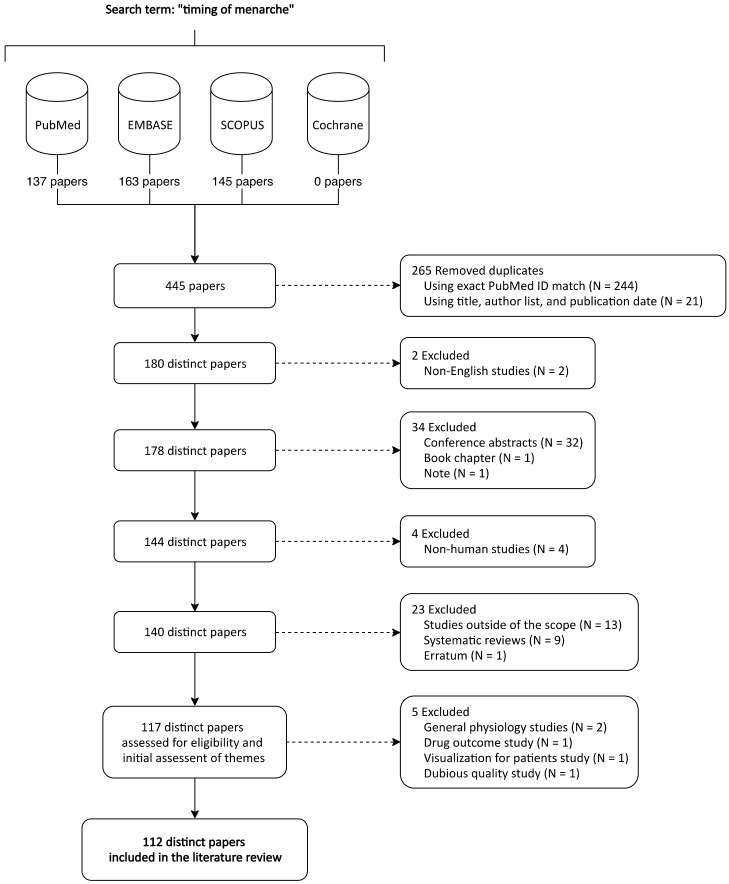
PRISMA flow diagram of study selection process. Initially 445 studies were identified. The review process resulted in 112 studies included in this systematic review.

**Figure 2 ijerph-17-01703-f002:**
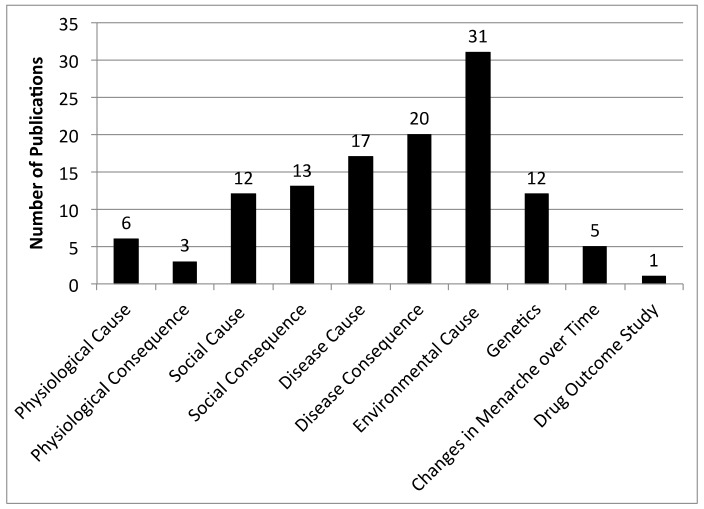
Breakdown of timing of menarche articles by themes identified. The 117 articles on timing of menarche were categorized into themes delineating articles on causes vs. consequences and other factors (e.g., genetics) described in the articles.

**Figure 3 ijerph-17-01703-f003:**
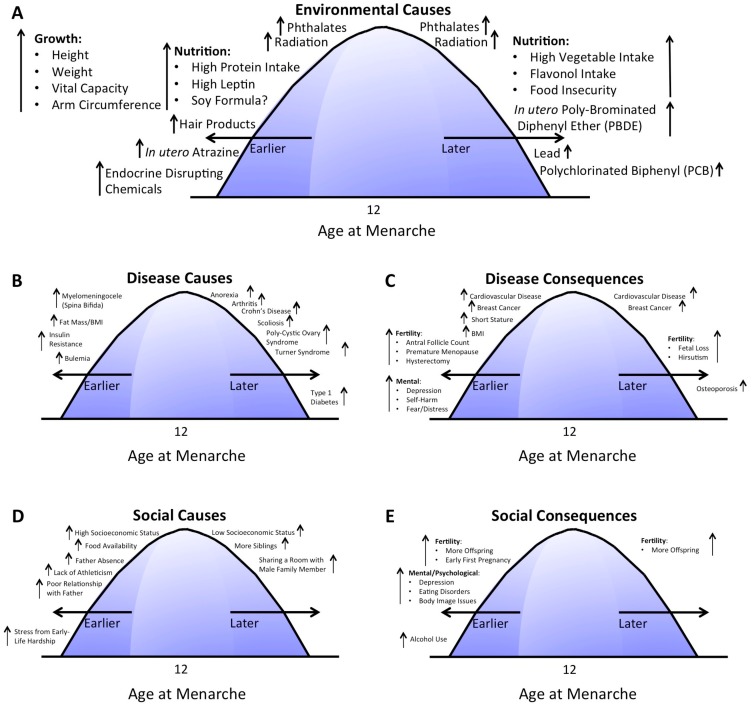
The causes and consequences of perturbations in timing of menarche. The environmental causes of either early or late menarche are shown in (**A**), followed by disease causes (**B**), and consequences (**C**), and social causes (**D**) and consequences (**E**). Mental health and psychological conditions were categorized as either diseases or social factors depending on the specific finding and research article. The arrows (↑) indicate that increases in that factor correspond to either early or late menarche (depending on the side of the bell distribution it is shown on). The question mark (?) that appears near “soy formula” indicates that there is conflicting evidence regarding whether or not soy formula has an effect on the timing of menarche.

**Figure 4 ijerph-17-01703-f004:**
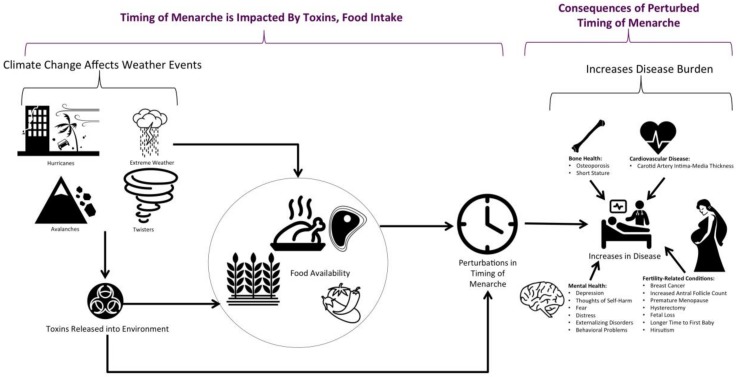
A conceptual schema illustrating how climate change could impact timing of menarche and increase disease burden. Climate change could impact weather events and thereby increase the toxins or toxicants released into the environment, affecting food and crops. This would impact crop availability and food intake and perturb timing of menarche. These perturbations in menarche could increase disease burden with regards to bone health, cardiovascular disease, mental health, and fertility-related conditions.

**Table 1 ijerph-17-01703-t001:** Diseases that Either Cause or Result from Discrepancies in Timing of Menarche.

Disease	Early or Late Menarche	Reference
**Disease Causes for Perturbations in Age at Menarche**		
Myelomingocele (a type of spina bifida)	Early	[62]
Fat mass	Early	[63]
Inverse association with BMI (i.e., heavier BMI - earlier menarche)	Early	[64,65]
Metabolic syndrome	Early	[66]
Insulin resistance	Early	[67]
**Not** Associated with Age at Menarche		
Birth weight	NA	[69]
Pre-term birth	NA	[70]
Small for gestational age	NA	[71]
Congenital adrenal hyperplasia	NA	[72]
Type 1 diabetes	Late	[64]
Anorexia	Late	[76]
Scoliosis	Late	[77]
Turner syndrome (one X chromosome)	Late	[78]
Polycystic ovarian syndrome (PCOS)	Late	[79]
Juvenile rheumatoid arthritis	Late	[80,81]
Crohn’s disease	Late	[82]
**Disease Consequences of Perturbations in Age at Menarche (i.e., resulting from delay or early menarche)**		
Short stature	Early	[86]
Elevated BMI	Early	[87]
**Mental Health Conditions:**		
Depression	Early	[90]
Thoughts of self-harm (but not suicidal)	Early	[91]
Fear	Early	[92]
Distress	Early	[92]
Externalizing disorders	Early	[92]
Behavioral problems	Early	[93]
**Cardiovascular disease:**	Early	[94,95,96]
Carotid artery intima-media thickness	Early	[95]
**Fertility-Related Conditions:**		
Increased Antral follicle count (typically a sign of increased fertility)	Early	[87]
Premature menopause	Early	[88]
Hysterectomy	Early	[89]
Breast Cancer	Early	[97,98,100]
**Fertility-Related Conditions:**		
Breast Cancer	Late	[100]
Fetal loss	Late	[84]
Longer time to first baby	Late	[84]
Hirsutism (excessive body hair growth)	Late	[85]
**Bone Health Conditions:**		
Osteoporosis	Late	[83]

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
