# Peer review of "A Systematic Literature Review of Factors Affecting the Timing of Menarche: The Potential for Climate Change to Impact Women’s Health"

_ijerph, 2020, doi:10.3390/ijerph17051703_

Round 1

Reviewer 1 Report

Silvia and Mary assessed the ways that climate, and also climate change, can affect timing of menarche, and potential consequences that this could have on women's health in this systematic review. The following corrections shall be made before further review.

  1. The introduction part is too general and it should concise with more scientific and statistical data.
  2. There are many mistakes which are appeared in the figures should be corrected
  3. How the authors considered/reviewed/excluded the paper related “timing of menarche” from the low income countries
  4. Author has used the search word “timing of menarche” with a valid reason but used “age at menarche” throughout the paper. Would you please change or explain the reason behind using different sentence?
  5. Exclude the information related to search engine from the methodology.
  6. The methodology part should be edited well for more clarity.
  7. Result section is too clumsy and repeated. This section should be edited well.
  8. The reason behind not restricting the number of years of publication should be explained.
  9. The number of papers reviewed is repeatedly used the paper and it should be avoided. Try to avoid the word “we” throughout the manuscript.  

Reviewer 2 Report

It is an important and well conducted literature review. Some comments that could improve the paper are:

- In the Introduction: The average age of menarche has declined in industrialized countries, as authors wrote. But there is also evidence of a decrease in the age of menarche in several developing countries.

- I think that many details of section 3.1 "Systematic Review of Literature" could be removed because they are repeated in Figure 1.

- Figure 1 is very clear, but I suggest reducing the explanation in the title of the figure (lines 152 – 156).

- There were 12 articles grouped in “genetics” (line 161). However, in the section 2.2 “Identification of themes” this theme does not appear.

-Data for Figure 2 are repeated in the text. They should be presented only once.

- Figure 3 is very clear, but in the environment causes there are several symbols # that make reading difficult.

- In the Introduction says that this review includes environmental, physiological, sociological, disease-related and genetic contributions to the age of menarche. When I read this, I wondered why psychological factors were omitted.  Then, in the Figure 3 I saw that they were considered as social factors. Therefore, I suggest changing “social” to “psychosocial”.

- I suggest shortening the explanation in Figure 4 (lines 390-397).

- The Discussion section is too long, and in many cases, it moves away from the issue of the timing of menarche. I recommend limiting the discussion to the objective of the paper.

Reviewer 3 Report

Authors have assessed  A Systematic Literature Review on Factors Affecting 2 the Timing of Menarche: Potential for Climate Change to Impact Women's Health. Authors have done a good job. Illustrations are really impressive and informative. Review have tried to cover all the area of climate change impacting women health. I have no minor or major concern. Manuscript can be accepted in present form. 

Round 2

Reviewer 1 Report

The authors addressed all the comments raised.